# Differential interactions of ToLCNDV with different betasatellites reveal complex viral dynamics in *N. benthamiana*

**Zafar Iqbal**[1☻*], **Muhammad Shafiq**[2☻*], **Sajed Ali**[2], **Mudassar Fareed Awan**[2], **Muhammad Farhan Sarwar**[2], **Imran Amin**[3], **Muhammad Shafiq Shahid**[4], **Rob W. Briddon**[3]

**1** Central Laboratories, King Faisal University, Al-Ahsa, Saudi Arabia, **2** Department of Biotechnology, University of Management and Technology, Sialkot Campus, Sialkot, Pakistan, **3** Agricultural Biotechnology Division, National Institute for Biotechnology and Genetic Engineering, Faisalabad, Pakistan, **4** Department of Plant Sciences, College of Agricultural and Marine Sciences, Sultan Qaboos University, Al-Khoud, Muscat, Oman

☻ These authors contributed equally to this work.
* zafar@kfu.edu.sa, zafariqbal2009@gmail.com (ZI); shafiq.4721@gmail.com (MS)

## Abstract

Tomato leaf curl New Delhi virus (ToLCNDV), a bipartite begomovirus prevalent in Old World and major cotton-growing regions of Pakistan, has increasingly been found associated with diverse betasatellites. Although betasatellites, small circular DNA satellites, are typically associated with monopartite begomoviruses, they are known to enhance disease severity (pathogenicity), increase viral DNA accumulation, and expand virus host ranges. This study investigated the interaction between ToLCNDV and three widely distributed betasatellites – cotton leaf curl Multan betasatellite strain Multan (Mβ), cotton leaf curl Multan betasatellite strain Burewala (Bβ) and tobacco leaf curl betasatellite (Tbβ) – in *Nicotiana benthamiana* plants, focusing on the potential emergence of novel viral combinations in cotton-growing regions. Infectious clones of ToLCNDV (either DNA-A [TA] alone or with DNA-B [TB]) were co-inoculated with each betasatellite clone. Results revealed intriguing complexity and variability in interactions: betasatellite with TA/TB affected TB accumulation, suggesting a competition between them, while TA levels increased only in the presence of TB, not apparently with Bβ. Interestingly, Tβ accumulated to the higher levels in plants, followed by Bβ and Mβ, highlighting betasatellite-specific interactions. These findings suggest that ToLCNDV-betasatellite interactions are more intricate than simple antagonism. The co-occurence of ToLCNDV with diverse betasatellites in cotton-growing regions of Pakistan increases the likelihood of the emergence of novel and potentially more pathogenic viral combinations. These intricate interactions have significant implications for understanding the dynamics of CLCuD disease.

**Data availability statement:** All the data related to this work is available in the manuscript and supplementary data. The research presented in this paper was originally conducted for the Ph.D. thesis of MS. This content has not been published elsewhere but a part of Higher education Commission (HEC) Pakistan's database.

**Funding:** The publication of this work was supported by the Deanship of Scientific Research (DSR), Vice Presidency for Graduate Studies and Scientific Research, King Faisal University, Saudi Arabia (KFU252251). The funder has no role in study design, data collection and analysis, decision to publish, or preparation of the manuscript.

**Competing interests:** The authors have declared that no competing interests exist.

## 1. Introduction

Tomato leaf curl New Delhi virus (ToLCNDV) is a highly widespread, prevalent and damaging bipartite begomovirus (family *Geminiviridae*), posing a significant threat to tomato production across three continents, including Asia, Africa, and Europe. It is a major impediment to agricultural productivity, particularly tomatoes, in South and Southeast Asia [1–3]. In recent years, it has expanded its reach from Asia to the Mediterranean region, causing concern in North Africa [4] and Southern Europe [5–7]. ToLCNDV infection leads to stunted growth and severe leaf curling in tomato plants and most plants it infects. ToLCNDV is a whitefly (*Bemisia tabaci*) transmitted bipartite begomovirus with two genomic components, DNA-A (TA) and DNA-B (TB), encoding eight genes. These genes are transcribed in both directions, virion-sense and complementary-sense, on both components. TA harbors genes for the coat protein (CP), pre-coat protein (AV2), and replication-associated protein (Rep), transcriptional activation (TrAP), replication enhancement (REn), and AC4. While TB encodes two proteins, the nuclear shuttle protein (NSP) and movement protein (MP), responsible for *in planta* movement. Prior studies have comprehensively characterized the functions of these genes and their crucial role in enabling successful infection by geminiviruses [8,9].

Most tomato infecting, single-stranded DNA begomoviruses from the Old World (OW) have monopartite genome and are associated with betasatellites. These small, circular, and single-stranded DNA molecules (genus *Betasatellite*, family *Tolecusatellitidae*) depend on their helper virus for replication, encapsidation, transmission, and *in planta* movement, and are in many cases, symptom determining [10–13]. Since their discovery in 1999, over 1300 isolates have been identified from 37 countries. Betasatellites typically harbor a single functional gene (βC1) on complementary-sense strand, which performs all the functions of betasatellite. There are, however, some exceptions, where another gene, referred to as βV1, is encoded on virion-sense strand and plays a critical role in infection, inducing mosaic and chlorosis symptoms [14]. Additionally, βV1 can elicit hypersensitive response (HR)-type cell death associated with reactive oxygen species (ROS) accumulation. Furthermore, βV1 can interact with the REn of helper virus [15]. On the other hand, βC1 performs diverse functions, including regulation of host microRNA levels [16], suppression of host defense at transcriptional gene silencing (TGS) and post transcriptional gene silencing (PTGS) [17,18], disrupt structure and functions of chloroplast [19], binds to DNA and RNA [20], subvert antiviral defense by interacting with extrinsic protein of Photosystem II (PsbP) and SNF1-related protein kinase (snRK1) [21,22], prevent degradation of viral protein by inhibiting the ubiquitin conjugase E2 which is involved in proteasome pathway [23], exhibits ATP hydrolysis activity that influences its DNA-binding activity and viral pathogenesis [24], and even suppresses the plant's jasmonic acid production [25].

Though bipartite begomoviruses are typically not associated with betasatellites, rare exception exists, including mung bean yellow mosaic India virus [26–28] and tomato yellow leaf curl Thailand virus [29]. Nonetheless, ToLCNDV has frequently been found associated with different betasatellites under natural conditions in

Pakistan and India. It has been found associated with cotton leaf curl Multan betasatellite strain Multan (Mβ), cotton leaf curl Multan betasatellite strain Burewala (Bβ) [30,31], chilli leaf curl betasatellite [32,33], tomato yellow leaf curl Thailand betasatellite [34], papaya leaf curl betasatellite [35], tomato leaf curl Jodeybpur betasatellite [36], tobacco leaf curl betasatellite (Tbβ) [37], and tomato leaf curl Bangladesh betasatellite [38] (S1 Table in S1 File). Additionally, ToLCNDV has been shown to trans-replicate different betasatellites in *N. benthamiana* plants under experimental conditions, including Mβ [39,40], tomato yellow leaf curl Thailand betasatellite, tomato leaf curl Patna betasatellite, radish leaf curl betasatellite, and tomato leaf curl Joydebpur betasatellite [41], and tomato leaf curl Bangladesh betasatellite [42].

Cotton leaf curl disease (CLCuD), a devastating disease caused by begomoviruses and their associated betasatellites, first emerged in Punjab in the 1960s [43]. After a period of localized outbreaks, a severe epidemic swept through Punjab in the 1990s, causing substantial economic losses reaching up to US$ 5 billion [44]. The emergence of resistant cotton varieties initially curbed the disease, but this resistance was short-lived, leading to a second epidemic in the early 2000s [45]. The first epidemic was primarily associated with Cotton leaf curl Multan virus (CLCuMuV) and Cotton leaf curl Kokhran virus (CLCuKoV), while the second was dominated by a recombinant CLCuKoV strain (CLCuKoV-Bur) [46]. In Sindh and northwestern India, greater begomovirus diversity persists, with CLCuKoV-Bur co-existing with other virus species. Recent reports indicate a resurgence of begomoviruses associated with the first epidemic of CLCuD in Pakistan [47] and India [48]. Additionally, ToLCNDV has also been reported in CLCuD-infected cotton in Pakistan [49]. A recent study [50] reported the presence of a novel and distinct CLCuMB lineage in Northwest India, alongside three alphasatellite species associated with CLCuD. This discovery underscores the ongoing evolution and diversification of the CLCuD disease complex in the region, particularly in the context of the widespread presence of ToLCNDV, which may further contribute to the emergence of novel and potentially more virulent viral combination.

Previous research using gene mutations in cotton leaf curl Kokhran virus and ToLCNDV [39,51] revealed the intricate interplay between betasatellites and gene mutants of their helper viruses, extending beyond just trans-replication and movement. While interaction between monopartite begomoviruses and their associated betasatellites have been extensively studied, knowledge about their interaction with bipartite begomoviruses remained an enigma. This complex interaction between ToLCNDV and betasatellites significantly worsens disease severity, yield losses, and economic hardship for farmers [38]. As these three betasatellites are frequently found associated with cotton infecting begomoviruses where ToLCNDV is also present, so there is a high potential for interaction that could lead to the emergence of a new viral combination with severe negative impact on crops. Therefore, understanding this intricate relationship is crucial for developing effective control strategies. This study aimed to investigate the interaction between ToLCNDV and diverse betasatellites.

## 2. Materials and methods

### 2.1. Origins of ToLCNDV and betasatellite clones and primers used in the study

Infectious binary vector constructs of TA (GenBank accession number U15015), TB (GenBank accession number U15016) [1], Mβ (GenBank accession number AJ292769) [13], Bβ (GenBank accession number AM774307) [46], and Tbβ (GenBank accession number AM922485) [26] have been described earlier. These constructs were electroporated into Agrobacterium tumefaciens (GV-3101) for inoculation in Nicotiana benthamiana plants.

To design primers for real time PCR and southern blot probe, sequences of TA, TB, Mβ, Bβ and Tbβ were retrieved from NCBI GenBank (https://www.ncbi.nlm.nih.gov/; accessed on December 2015). Each set of retrieved sequences were aligned in MEGA using Clustal X2 [52]. Then different set of oligonucleotide primers were designed on the highly conserved regions identified in a multiple sequence aligned files (S2 Table; S1 Fig in S1 File).

### 2.2. Sequence alignment and SCR analysis

The nucleotide sequences of all three betasatellites were retrieved from NCBI (https://www.ncbi.nlm.nih.gov/) and aligned using MEGA11 for multiple sequence comparison. Two datasets were then generated: one containing the coding

sequence (CDS) of βC1 and the other representing the satellite conserved region (SCR) and some of its upstream region. The βC1 sequences were translated into amino acids to evaluate their similarity at the protein level. Additionally, we conducted conserved repetitive motif analysis within the SCR of the three betasatellites using NovoPro's repeat sequence finder (https://www.novoprolabs.com/tools/repeats-sequences-finder) to identify shared regulatory elements.

## 2.3. Agroinoculation of *N. benthamiana* plants

A. tumefaciens (GV3101) cultures were grown in LB medium supplemented with rifampicin (25 µg.mL$^{-1}$), kanamycin (50 µg.mL$^{-1}$), and tetracycline (50 µg.mL$^{-1}$) at 28 °C for 48 h to an OD600 of 0.6–0.8. After centrifugation (5,000 × g for 15 min at 20°C), cells were resuspended in 10 mM MgCl$_2$ solution, adjusted to a uniform OD of 2.0. The Agrobacterium inoculums were further diluted to an OD of 1.0 and different inoculums were mixed to ensure that all plants receive equal concentration (OD = 0.3) of each construct. Finally, the inoculum was incubated for 1 h at room temperature containing 150 µg.mL$^{-1}$ acetosyringone. A. tumefaciens cultures harboring the infectious constructs were inoculated into N. benthamiana plants (6–8 leaf stage). A group of six control plants received mock inoculation with Agrobacterium inoculum containing only the binary vector without the construct. Two to three leaves per plant were infiltrated with 0.5–1 mL of inoculum using a sterile 1 mL syringe [53]. For each experiment, control (healthy) plants without Agroinfiltration were included and each treatment involved inoculating six plants, with the experiment being repeated three times. The plants were housed in a pest-free glasshouse and monitored closely for symptom development each day. At 25 days post-inoculation (dpi), young leaf samples (emerging after inoculation) were used for DNA extraction and subsequent analysis using quantitative real-time PCR (qPCR) and Southern blotting. Plants were photographed prior to the sample collection.

## 2.4. Southern blot hybridization and bands quantification

Genomic DNA was extracted from 100 mg of leaf tissue using the CTAB method [54]. This DNA, isolated from two individual plants confirmed positive for viral components by PCR, was then analyzed by Southern blotting. For each sample, 8 µg of DNA was electrophoresed on agarose gels (1.5%), transferred to nitrocellulose membranes, and hybridized with α32P-dCTP labelled probes, following standard protocols [55]. Specific probes were used for the detection of TA (628 bp fragment of the Rep Gene), TB (475 bp fragment of the MP gene), and respective βC1 gene of each betasatellites (S1 Table; S1 Fig in S1 File). Following hybridization, detection was performed using a phosphor screen scanned on a Pharos FxTM system and analyzed with Quantity one software (version 4.6.9). Southern blots were performed on experiment 3 plants.

Southern blot band intensities were quantified using ImageJ software [56]. For each blot, the positive control band (TA in TA-inoculated plants, TB in TA/TB-inoculated plants, and betasatellite in TA/betasatellite-inoculated plants) was set to 100%, and all other bands were quantified relative to this control.

## 2.5. Quantitative analysis of the viral components

Genomic DNA from each plant was subjected to qPCR using the Chromo4TM PCR Detection System (Bio-Rad). The SYBR Green PCR Master Mix (ThermoFisher Scientific, Germany) was used for the assay, with specific primers designed (S1 Table in S1 File) to quantify each target viral component. Both reaction mixture and PCR conditions have already been described [57,58]. Standard curves were generated using ten-fold serial dilutions of standards (plasmid DNA spiked with N. benthamiana DNA) and melt curve analysis (S2 Fig in S1 File) was performed as outlined earlier [57]. The expression of each studied component was normalized to the 18S RNA housekeeping gene. Extracted DNA from the leaves of healthy N. benthamiana and mock inoculated plants were used as a negative control in qPCR assay. qPCR was performed on three biological replicates (comprising of three independent experiments), each with three technical replicates. Data from one representative biological replicate (Experiment 3) is presented for clarity and conciseness.

## 2.6. Standard curve analysis

Following each qPCR run, standard curve analysis was performed for TA, TB, and betasatellites amplicons using linear regression. This analysis plotted threshold cycles (Ct) values plotted against the input DNA amount. All the curves for each studied viral component demonstrated a linear relationship, and each sample showed a single peak, confirming the amplification of a single amplicon (S2 Fig in S1 File).

## 2.7. Molecular docking

To investigate the interaction between the Rep protein and target DNAs, molecular docking studies were performed. The three-dimensional (3D) structure of the Rep protein and DNA sequences representing the conserved region (CR) of TB, and the SCR of three betasatellites were constructed in silico. All molecular structures were prepared and saved in Protein Data Bank (.pdb) format using BIOVIA Discovery Studio Visualizer (v. 2024). Subsequently, protein-nucleic acid docking was carried out using the HDOCK server (http://hdock.phys.hust.edu.cn). The HDOCK server was selected for its specialized capability in performing protein-nucleic acid molecular docking analysis [59]. The respective .pdb files of the Rep protein and the DNA molecules were uploaded and processed through the server.

The resulting docked complex files, generated by the HDOCK server in.pdb format, were then analyzed using BIOVIA Discovery Studio Visualizer. Post-processing of these results included visualization of the docked structures, identification of interacting atoms, and figure generation.

## 2.8. Statistical analysis

The analysis of the qPCR data was performed using OriginPro software (version 10.1). Each treatment group consisted of three replicates (n = 3), and the mean values were employed for the analysis. One-way analysis of variance (ANOVA) was conducted to assess differences between multiple groups, followed by Tukey's post-hoc test for pairwise comparisons at $p \leq 0.05$.

# 3. Results

## 3.1. Sequence analysis of βC1 and SCR of the betasatellites

Pairwise sequence analysis of the βC1 protein region revealed that Tbβ exhibited a sequence identity of 15% and 15.8% with Bβ and Mβ, respectively, indicating a relatively distant evolutionary relationship. In contrast, Bβ and Mβ shared a significantly higher sequence identity of 96.1%, suggesting a closer evolutionary relationship between these two betasatellites (S3A Fig in S1 File).

Similarly, the SCR of Tbβ was more dissimilar to Bβ and Mβ compared to the similarity between Bβ and Mβ. While the SCR of Bβ and Mβ shared a significantly higher sequence identity, suggesting a conserved region important for their function or interaction with other viral components. (S3B Fig in S1 File). Analysis of the satellite conserved region (SCR) and its upstream region in TA, TB, and the three betasatellites revealed variations in conserved motifs and differing levels of similarity to TA. TB exhibited the greatest similarity to TA, sharing the first motif (5'-GGTGTCT-3') and differing only in the second (5'-GGCGTCT-3' vs. 5'-GGTGTCT-3' in TA). Bβ's first motif (5'-GGTGTTC-3') showed a slight variation, while its second (5'-GGTGTGT-3') diverged further. Tbβ's motifs (5'-GGCGTCA-3' and 5'-GGCGTAA-3') displayed moderate divergence from TA. Notably, Tbβ and Bβ shared a striking proximity in the location of their first motifs, hinting at a possible shared functional significance. Mβ was the most dissimilar, with both motifs (5'-GGTGATC-3') distinct from TA. These motif variations likely contribute to observed functional differences, including Bβ's superior performance over Mβ, despite 96.1% βC1 similarity.

## 3.2. Inoculation of ToLCNDV alone and with different betasatellites

N. benthamiana plants inoculated with only TA resulted in milder disease symptoms (100% infection) compared to those inoculated with both TA and TB (Fig 1A). These mild symptoms, characterized by upward leaf curling, emerged on newly developing leaves at 9 dpi and remained the same until the sampling time at 25 dpi.

In contrast, N. benthamiana inoculated with both TA and TB led to severe infection in all the inoculated plants. Notably, there was no difference in latent period and the first visible symptoms were upward leaf curling (Fig 1B; Table 1). Nonetheless over time, the severity of the symptoms intensified, leading to pronounced upward leaf curling.

Importantly, all non-inoculated control (healthy) plants and mock-inoculated plants remained symptomless throughout the experiment and were comparable to each other (Fig 1I, 1J).

The investigation into the co-infection effects of ToLCNDV and three associated betasatellites on N. benthamiana plants revealed a fascinating interplay between these pathogens. Co-inoculationof TA and TB along with Mβ resulted in a 100% infection rate in N. benthamiana plants, but with characteristic difference in symptoms. Interestingly, the primary symptom observed was a downward leaf curl, contrasting with the upward curling seen in TA/TB co-infection (Fig 1C). Notably, the symptoms onset was delayed, appearing at 10 dpi compared to 9 dpi observed with TA/TB (Table 1). This infection intensified over time, resulting in severe downward leaf curling and leaf crumpling.

Interestingly, co-inoculation of just TA along with Mβ also led to a 100% infection, but milder than TA/TB/Mβ inoculated plants (Fig 1D). Again, symptoms onset was slightly delayed (10 days compared to 9 days without the betasatellite; Table 1). The observed symptoms in this case were mild downward leaf curling of the edges of the leaves, forming inverted cup shape like structure with vein swelling (Fig 1D). This phenotype remained consistent throughout the experiment period.

In stark contrast to TA/Mβ inoculated plants, all the N. benthamiana plants inoculated with TA/Bβ or TA/TB/Bβ displayed symptoms more reminiscent of TA/TB than those observed with a betasatellite. These symptoms included upward leaf curling and vein swelling (Fig 1E, 1F).

Notably, N. benthaimana plants co-inoculated with TA/TB and Tbβ exhibited a substantially distinct disease phenotype compared to those co-inoculated with either Mβ or Bβ. Unlike the characteristic upward curling and crumpling seen with the latter two, Tbβ co-inoculated plants exhibited a distinctive downward curling (invert cupping) of leaves (Fig 1G, 1H). This cupping effect was accompanied by less crumpling and visible swelling of the veins. Interestingly, the latent period remained consistent at 10 days post-inoculation (dpi) regardless of Tbβ presence. However, a crucial difference emerged in the severity of the symptoms between plants infected with TA/TB alone and those TA/TB and Tbβ. In the absence of

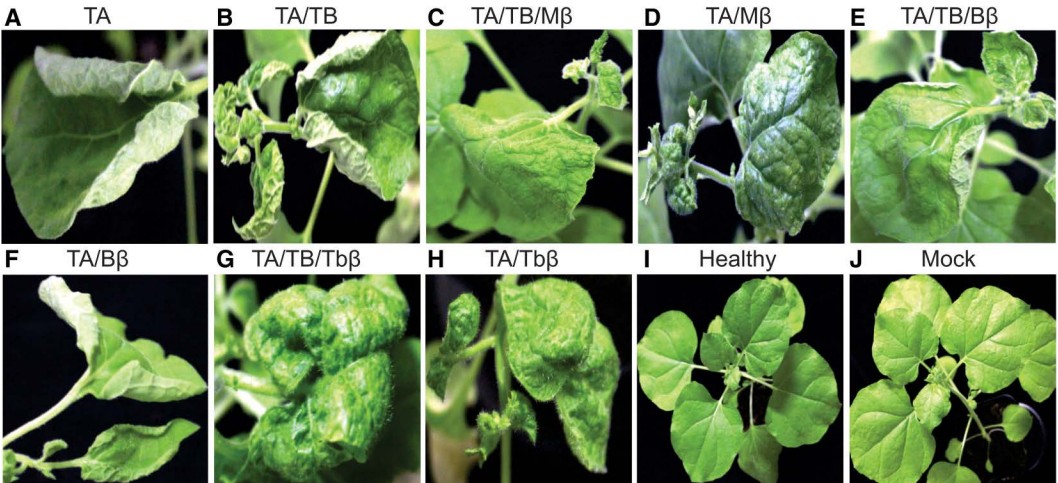

**Fig 1. Impact of different betasatellites on symptom development in ToLCNDV-inoculated *N. benthamiana* plants.** Plants were inoculated with TA (A), TA and TB (B), TA, TB and Mβ (C), TA and Mβ (D), TA, TB and Bβ (E), TA and Bβ (F), TA, TB and Tbβ (G), TA and Tbβ (H, non-inoculated (Healthy; I) and mock (J). Abbreviations used in the figure are ToLCNDV DNA-A (TA), DNA-B (TB), cotton leaf curl Multan betasatellite (Mβ), cotton leaf curl Multan betasatellite strain Burewala (Bβ), and tobacco leaf curl betasatellite (Tbβ).

TB, the symptoms were a bit milder. Furthermore, the presence of Tbβ appeared to influence the progression of symptoms over time.

### 3.3. Southern blot detection and quantification of viral components

Southern blot band intensities were quantified for each viral component, ensuring that band signals remained within the linear range of detection and exhibited minimal background noise. Within each blot, the corresponding positive control band (TA in TA-inoculated plants, TB in TA/TB-inoculated plants, and betasatellite in TA/betasatellite-inoculated plants) was normalized to 1.0 (100%), and all other bands were quantified relative to this control, and results are depicted in ratios (S4 Fig in S1 File). Notably, TA DNA was consistently detected in all inoculated plants, regardless of whether they were co-infected with TB or with different betasatellites (Mβ, Bβ, or Tbβ). TA DNA levels were slightly higher in plants co-inoculated with TB (1.38) compared to plants inoculated with TA alone (1.0). This indicates that TA establishes itself within the plant system. However, co-inoculation with any of the betasatellite further elevated TA levels, suggesting that these satellite molecules may influence TA accumulation or persistence within the host plant (Fig 2). Among the different co-infections involving betasatellites, Tbβ co-inoculation resulted in the highest accumulation of TA DNA (2.08–2.43), followed by Mβ (1.46–1.64) and Bβ (1.39–1.57), respectively (S4A Fig in S1 File). This finding suggests that Tbβ might be particularly adept at enhancing TA replication or persistence within the host than Mβ and Bβ (Fig 2). TA ssDNA quantification showed the similar results with the highest levels in Tbβ co-inoculations, followed by Mβ, and the lowest in Bβ co-inoculations. This suggests varying influences of the betasatellites on TA replication, with Tbβ exhibiting the strongest synergistic interaction (S4B Fig in S1 File).

In Southern blot analysis, TB was successfully detected in plants inoculated with either TA/TB or in combination with any of the betasatellites (TA/TB/betasatellite) (Fig 3). The TB was accumulated to a highest level in TA/TB inoculated plants (1.0) than in presence of any betasatellite; TA/TB/Mβ (0.59), TA/TB/Bβ (0.74), and TA/TB/Tbβ (0.96) (S4C Fig in S1 File). Importantly, none of the control plants reveal the detection of either TA nor TB (Fig 3). This negative result in control plants strengthens the validity of the experiment and confirms the specificity of the probe used. It is noteworthy that the Southern blot for TB detection required an extended exposure time to the phosphor imager to yield the visible results.

**Table 1. Evaluation of ToLCNDV infectivity in *N. benthamiana* plants co-inoculated with various betasatellites.**

| Inoculum | Infectivity | | | | | | Symptoms$ | Symptoms Severity Score | Latent period (days) |
|---|---|---|---|---|---|---|---|---|---|
| | Diagnostics by qPCR (plants infected/ inoculated)* | | | Southern | | | | | |
| | | | | TA | TB | Beta | | | |
| | | | | TA | TB | Beta | | | |
| TA | 18/18 | -- | -- | + | -- | -- | MULC | + | 9 |
| TA/TB | 18/18 | 18/18 | -- | + | + | -- | SULC,VT | ++ | 9 |
| TA/TB/Mβ | 18/18 | 18/18 | 18/18 | + | + | + | SDLC,VT | +++ | 10 |
| TA/Mβ | 18/18 | -- | 18/18 | + | -- | + | MULC,VT | +++ | 10 |
| TA/TB/Bβ | 18/18 | 18/18 | 18/18 | + | + | + | SULC,VT | ++ | 10 |
| TA/Bβ | 18/18 | -- | 18/18 | + | -- | + | MULC | + | 10 |
| TA/TB/Tbβ | 18/18 | 18/18 | 18/18 | + | + | + | SDLC,VT | +++ | 10 |
| TA/Tbβ | 18/18 | -- | 18/18 | + | -- | + | SDLC,VT | +++ | 10 |

*Cumulative average of three independent experiments. Each experiment comprised 6 plants per inoculum type.

$Symptoms are denoted are mild upward leaf curling (MULC), severe upward leaf curling (SULC), severe downward leaf curling (SDLC), and vein thickening (VT). Two consecutive dash lines (--) represents viral component was not used in the inoculum. Abbreviations used in the table are ToLCNDV DNA-A (TA), DNA-B (TB), cotton leaf curl Multan betasatellite (Mβ), cotton leaf curl Multan betasatellite strain Burewala (Bβ), and tobacco leaf curl betasatellite (Tbβ).

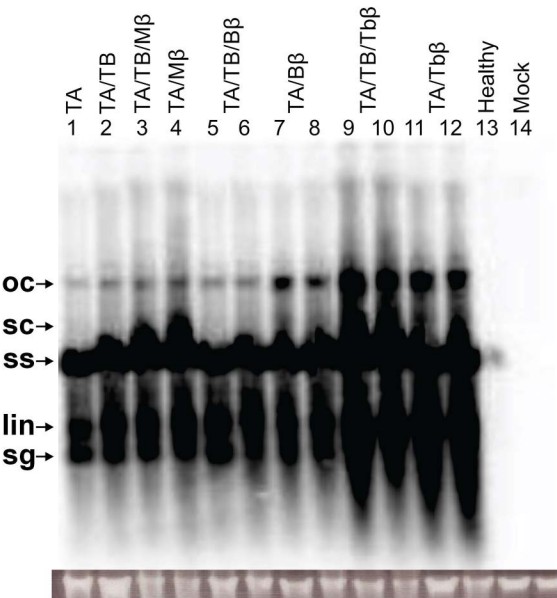

**Fig 2. Southern blot for the detection of TA in agro-inoculated *N. benthamiana* plants.** DNA samples from leaf extracts were run on agarose gel: TA (lane 1), TA and TB (lanes 2), TA, TB and Mβ (lanes 3), TA and Mβ (lanes 4), TA, TB and Bβ (lanes 5 and 6), TA and Bβ (lanes 7 and 8), TA, TB, and Tbβ (Lanes 9 and 10), TA and Tbβ (Lanes 11 and 12), non-inoculated healthy plant (lane 13) and a mock inoculated plant (lane 14). The agarose gel image (shown below the Southern blot) confirms equal loading of DNA samples. Viral DNA forms are labelled as linear (lin), open-circular (oc), single-stranded (ss), super-coiled (sc), and sub-genomics (sg). Abbreviations used in the figure are ToLCNDV DNA-A (TA), DNA-B (TB), cotton leaf curl Multan betasatellite (Mβ), cotton leaf curl Multan betasatellite strain Burewala (Bβ), and tobacco leaf curl betasatellite (Tbβ).

Further Southern blot analysis focused on the detection of specific betasatellites. Mβ was detected in plants co-inoculated with either TA or TA/TB. Mβ accumulation, as indicated by band intensity and replicative forms, was slightly higher in TA/TB/Mβ-inoculated plants (0.83–1.0) compared to TA/Mβ-inoculated plants (0.71–0.79) (Fig 4A; S4D Fig in S1 File). This higher accumulation suggests that the presence of TB might enhance the Mβ accumulation. In stark contrast to Mβ, the Southern blot analysis of Bβ and Tbβ (Figs 4B and 4C, respectively) revealed slightly higher accumulation. Plants co-inoculated with TA/Bβ or TA/TB/Bβ displayed comparable levels of betasatellites, ranging from 1.0 to 1.15. Likewise, Tbβ inoculated plants showed higher accumulation of Tbβ (1 to 1.23) compared to the other two betasatellites. These observations suggested that presence or absence of TB did not significantly affect the accumulation levels of Bβ and Tbβ. Notably, DNA from plants inoculated with other betasatellites was included as controls in each gel. For instance, in the Mβ Southern blot, DNA from Bβ-inoculated plants was included. No cross-detection of betasatellites was observed, suggesting highly specific detection methods.

### 3.4. qPCR-mediated quantification of viral components

To ensure consistency in the results, we included data from plants analyzed using Southern blotting alongside other plants. Furthermore, qPCR assays incorporated multiple biological and technical replicates to enhance statistical robustness. Rigorous optimization of qPCR conditions, including primer concentrations and thermocycler settings, was performed to guarantee consistent and efficient amplification across all viral components. Melt curve analysis of the qPCR reactions revealed a single, distinct peak, confirming the amplification of a unique and specific product for each target. This observation indicates high specificity and the absence of non-specific amplification or primer-dimer formation. While significant variation in the levels of viral DNA components was observed across all plants, the mean and cumulative average of viral DNA and their associated betasatellites are mentioned in the folllwing results.

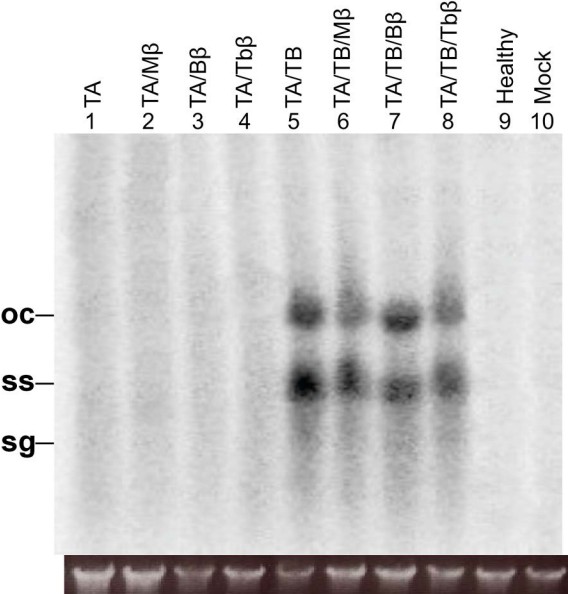

**Fig 3. Southern blot for the detection of TB in agro-inoculated *N. benthamiana* plants.** The DNA samples loaded onto the gel came from the leaves of plants that inoculated with TA (lanes 1), TA and Mβ (lane 2), TA and Bβ (lane 3), TA and Tbβ (lane 4), TA and TB (lane 5), TA, TB and Mβ (lane 6), TA, TB and Bβ (lane 7), and TA, TB and Tbβ (lane 8), and of control plants, healthy (non-inoculated) (lane 9) and a mock plant (lane 10). The agarose gel image (shown below the Southern blot) confirms equal loading of DNA samples. Abbreviations used in the figure are ToLCNDV DNA-A (TA), DNA-B (TB), cotton leaf curl Multan betasatellite (Mβ), cotton leaf curl Multan betasatellite strain Burewala (Bβ), and tobacco leaf curl betasatellite (Tbβ).

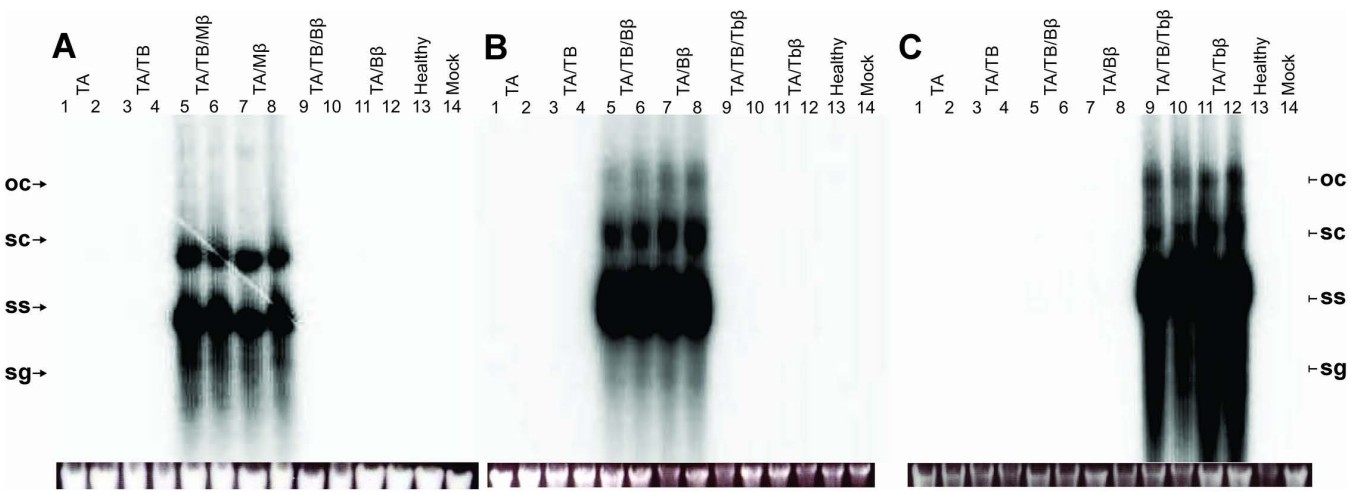

**Fig 4. Southern blot Mβ (A), Bβ (B) and Tbβ (C) in agro-inoculated *N. benthamiana* plants.** For the detection of Mβ (A), DNA samples extracted from leaves of plants inoculated with TA (lanes 1 and 2), TA and TB(3 and 4), TA, TB and Mβ (5 and 6), TA and Mβ (7 and 8),TA, TB and Bβ (9 and 10),TA and Bβ (11 and 12), healthy (non-inoculated) plant (13) and a mock inoculated plant (lane 14). The agarose gel images (shown below the Southern blot) confirm equal loading of DNA samples. Abbreviations used in the figure are ToLCNDV DNA-A (TA), DNA-B (TB), cotton leaf curl Multan betasatellite (Mβ), cotton leaf curl Multan betasatellite strain Burewala (Bβ), and tobacco leaf curl betasatellite (Tbβ).

In qPCR analysis, none of the viral components were detected in control plants, validating the specificity of qPCR and experimental setup. The mean value of the measured titers of TA in plants inoculated with TA/TB were insignificantly higher (0.0955 ng/μg of genomic DNA) compared to TA inouclated plants (0.0645 ng/μg of genomic DNA) (Fig 5A). The presence or absence of betasatellites significantly affect TA titers in TA/TB co-inoculated plants. Interestingly,TA titers were nearly doubled in plants inoculated with TA/betasatellites compared to those inoculated with TA/TB/Mβ. Notably, the highest TA accumulation (0.457 ng) was observed in plants inoculated with TA/TB/Tbβ (Fig 5A).

The TB DNA was not detected in control plants, and those inoculated with TA, TA/Mβ, TA/Bβ, and TA/Tbβ, validating the qPCR results (Fig 5B). Interestingly, plants inoculated with TA/TB/betasatellite led to a slightly higher TB accumulation than TA/TB plants. In TA/TB/Mβ, TA/TB/Bβ, and TA/TB/Tbβ plants, TB titers were slightly high, reaching a mean value of 0.00049, 0.000495, and 0.00052 ng/μg of genomic DNA, respectively, compared to 0.00047 ng/μg of genomic DNA in plants inoculated with TA/TB (Fig 5B). While our data showed slightly elevated TB titers, it's important to note that this increase was not statistically significant.

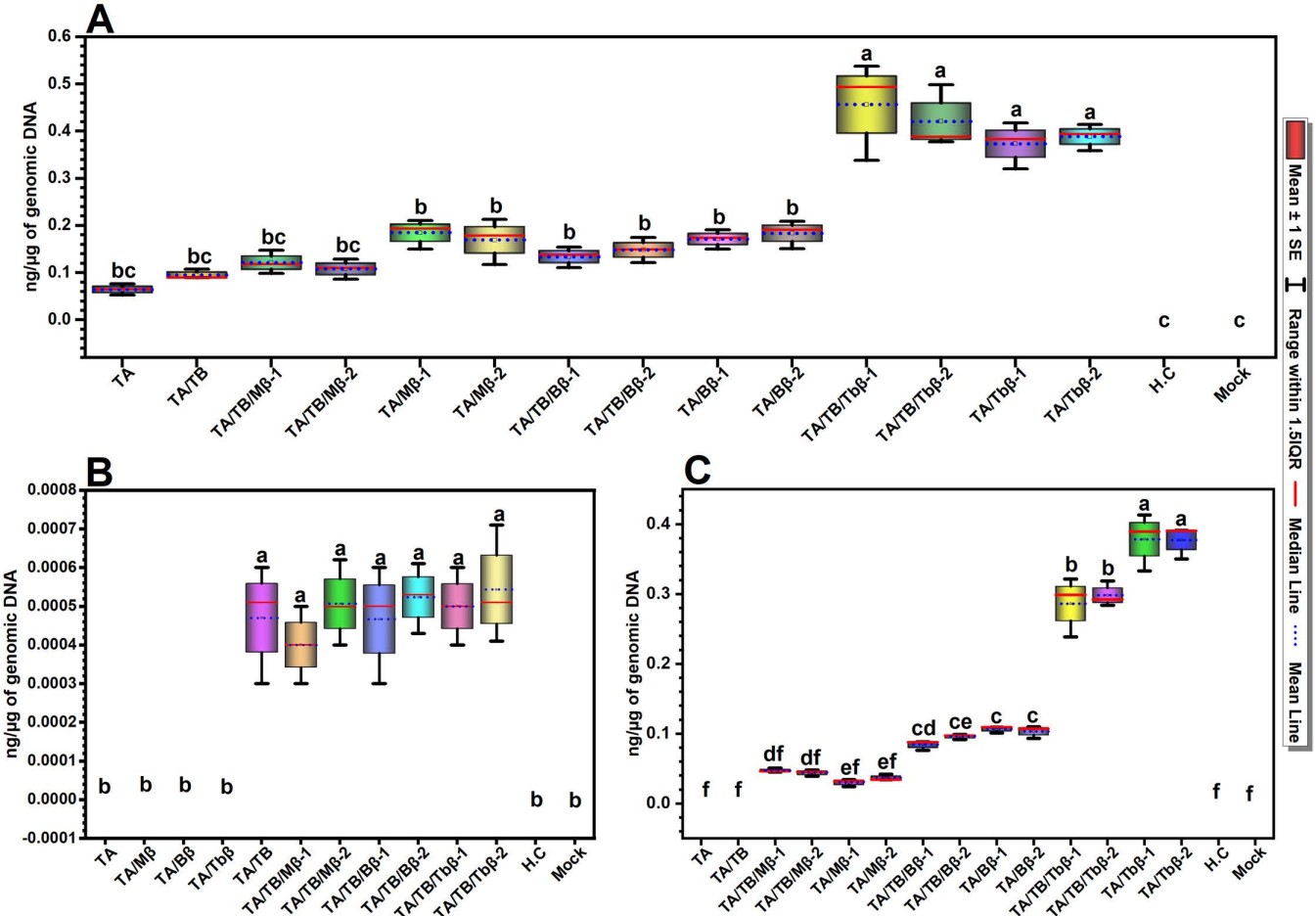

**Fig 5. Estimation of the DNA titers of TA (A), TB (B) and betasatellite (C) in the inoculated plants, control healthy (non-inoculated; H.C) and mock plants.** The experiment was repeated three times, with samples from each experiment analyzed in triplicate (technical repeats). The data presented in the figure represents the mean values obtained from three independent replicates (n = 3). Statistical analysis was performed, and groups denoted by different lowercase letters differ significantly from each other at a 5% significance level (p ≤ 0.05). Abbreviations used in the figure are ToLCNDV DNA-A (TA), DNA-B (TB), cotton leaf curl Multan betasatellite (Mβ), cotton leaf curl Multan betasatellite strain Burewala (Bβ), and tobacco leaf curl betasatellite (Tbβ). Median and mean are shown in solid red and dotted blue lines, respectively.

qPCR analysis revealed significant differences in betasatellite accumulation. Tbβ exhibited significantly higher titers than Mβ and Bβ in all combinations, reaching ~0.298 and 0.378 ng/µg of genomic DNA in TA/TB/Tbβ and TA/Tbβ plants, respectively. Mβ showed the lowest accumulation, with titers around 0.047 and 0.037 ng/µg of genomic DNA in TA/TB/Mβ and TA/Mβ plants, respectively. Bβ titers were significantly higher than Mβ, reaching 0.096 and 0.106 ng/µg of genomic DNA in TA/TB/Bβ and TA/Bβ plants, respectively (Fig 5C).

### 3.5. Molecular docking of rep with CR of TB and SCR of betasatellites

Molecular docking analysis revealed distinct binding affinities and interaction energies between the Rep protein and each DNA sequence (S3 Table in S1 File). The Rep-TA (−241.17 kJ.mol⁻¹) and Rep-Tbβ (−189.72 kJ.mol⁻¹) complexes exhibited the highest binding affinity (lowest binding energies), followed by Rep-TB (−184.60 kJ.mol⁻¹), Rep-Bβ (−184.11 kJ.mol⁻¹), and Rep-Mβ (−184.09 kJ.mol⁻¹), suggesting a hierarchy in Rep's binding preference. In addition, this trend was further supported by the intermolecular interactions, which consistently followed the same affinity ranking.

In a competitive biological context—where Rep mediates the replication of multiple viral components—these findings imply that Rep likely prioritizes binding to TA, Tbβ, and TB due to its stronger affinity, potentially outcompeting with other betasatellites with weaker interactions. Notably, while Bβ and Mβ shared the same binding site and energy on Rep, Bβ formed more stabilizing interactions and close proximity (S5 Fig in S1 File), which may explain its comparatively better maintenance over Mβ in viral replication.

### 4. Discussion

Bipartite begomoviruses are increasingly being detected in the field alongside betasatellites, underscoring the need of investigation [31,60,61]. Among the bipartite begomoviruses, ToLCNDV stands out for its exceptionally broad host range and frequent association with diverse betasatellites, making it an intriguing subject for investigation. Furthermore, its wide-spread presence in cotton-growing areas, coupled with the rapid spread of PeLCV (along with its cognate betasatellite, TbLCB), raises concerns about the potential emergence of novel and more virulent viral combinations. This epidemiological context underscores the need to investigate the potential interactions between ToLCNDV and these diverse betasatellites (CLCuMB, CLCuMB-Bu, and TbLCB) to assess their impact on disease severity. While interactions and effects of betasatellites with monopartite begomoviruses have been extensively studied [40,51,62,63], the impact of their association with bipartite counterparts remains largely unexplored. This study aims to fill this gap by investigating the interaction of betasatellites with a bipartite begomovirus.

While bipartite begomoviruses typically require both DNA-A and DNA-B for symptomatic infection in plants, DNA-A (TA) alone has been shown to infect *N. benthamiana*, albeit inconsistently, with reduced viral titters and diminished infection efficiency [13,39,64,65]. Also, such infections seem limited to *N. benthamiana* and a typically observed only after Agro-inoculation, not mechanical or biolistic inoculation methods [1]. Our findings align with previous reports, with the exception of TA alone exhibiting symptoms albeit with milder symptoms, similar peculiar phenomena also observed by Jyothsna et al. [31]. Conversely, two other studies [39,66] observe asymptomatic infections with TA alone, suggesting isolate specific variations. Intriguingly, TA's systemic movement without TB deviates from typical bipartite begomovirus transport. This could reflect an evolutionary adaptation, perhaps analogous to monopartite begomoviruses, where TA may have acquired DNA-B while retaining independent movement, providing a selective advantage. Alternatively, its autonomous movement might be context-dependent, possibly only related to *N. benthamiana*'s lack of RDR6, a key player in antiviral silencing [67]. The absence of RDR6 could enable TA to bypass typical constraints. This unique behavior also raises the possibility of TA forming new pathogenic combinations by associating with other monopartite begomoviruses, the DNA B of bipartite begomoviruses, or satellites, potentially creating recombinant strains with altered disease dynamics, pathogenicity, and host range [68]. Understanding these mechanisms provides insights into the evolutionary flexibility of pathogen transport, aiding the development of innovative antiviral strategies and improving management of viral infections in key crops.

While co-inoculation with cognate betasatellites typically boosts monopartite begomovirus DNA titers [13,63,69,70], this study observed an intriguing exception. All the betasatellites increased TA titer but it was highest in the presence of Tbβ (Fig 5A). This suggests a positive influence of betasatellites on TA [31]. As we found here, they [31] observed that TB presence reduced betasatellite titer while the betasatellite presence enhanced TB titer. Notably, our study found that plants infected with Tbβ and Bβ showed significantly higher TB titers compared to those infected with TA and TB. The precise reason for this apparent antagonism is far from a conclusion. Our molecular docking results provided potential mechanistic insights, showing Rep's preferential binding to Tbβ and TB over other betasatellites, which might explain higher TB's replication advantage. This binding hierarchy could reflect competition for virus TA-encoded Rep by both DNA-B and betasatellites for replication, or interference with movement functions. As TB encoded proteins [71,72] and the βC1 protein [73] have movement functions, suggests a potential redundancy in the viral movement mechanism. This redundancy could potentially lead to competition or interference between the two mechanisms. Further research is needed to elucidate the molecular basis of the apparent antagonism. Or we may speculate that the difference likely stems from variations in viral isolates/strains, experimental conditions/design, environmental conditions, and replication dynamics. Moreover, Mβ impaired ToLCNDV ability to maintain cotton leaf curl Multan alphasatellite in *N. benthamiana* plants [40], highlighting further complexities in the intricate interplay among begomovirus-betasatellite-alphasatellite.

Intriguingly, Tbβ induced more pronounced disease symptoms compared to Bβ and Mβ, with a significantly higher DNA titer (Fig 5), aligning with the earlier study showing higher betasatellite DNA levels enhanced symptoms severity, impaired plant performance, and lower seed yield [58]. This also imply that, beyond simple titre, Tbβ's specific interaction mechanisms with the host and ToLCNDV played a crucial role, underscoring the concept of betasatellite-specific interactions. The lower titers of Bβ and Mβ, despite their persistent association with CLCuD, are surprising. This disparity suggests that different betasatellites might employ distinct mechanisms to exploit the host environment and their cognate virus. For example, host factors can influence tissue-specific viral gene expression via interactions with viral gene promoter regions [74]. So, *N. benthamiana* environment appears to favor Tbβ's replication and its interaction with ToLCNDV, possibly due to elements like the G-box motif in the βC1 region of Tbβ, known to enhance betasatellite replication through interactions with host factors [75]. Furthermore, non-coding regions of satellite DNA, as identified by Li et al. [76] and Reddy et al. [77], are also critical for replication and symptoms induction. Further investigation of these non-coding regions in these complex interactions is warranted and could be the interest of futuristic studies. Our findings revealed unique interactions between ToLCNDV and even quite closely related betasatellites. While the underlying mechanisms require further study, several observations are noteworthy. First, TBβ, despite its soybean origin [26], impacted TA infections more strongly than either of the Mβ and Bβ, suggesting adaptation to *Solanaceous* plants. Second, the contrasting effects of the two CLCuMBs are surprising. Their key difference lies in a non-coding region of SCR encompassing approximately 110 nt fragment from tomato leaf curl betasatellite [78,79]. However, this difference is unlikely the primary driver of observed plant responses. Instead, the six amino acid differences within the βC1 protein of these clones are far more likely to be responsible [80], as all known betasatellite functions are attributed to the βC1 protein [63,81]. Furthermore, high-affinity interactions between Rep-binding sites in the betasatellite and Rep proteins may influence betasatellite titer levels [24,82]. Third, our docking analysis revealed a binding affinity hierarchy (Tbβ > Bβ > Mβ) mediated by differential bonding, suggesting Rep-binding site affinity influences betasatellite accumulation.

The intricate relationship between a bipartite begomovirus and its associated betasatellite extends far beyond a simple exchange of replication machinery. While likely targeting the same host cells for synchronized infection, betasatellite act as a master manipulator. It hijacks the begomovirus's replication machinery, diverting resources for its own replication and suppressing the host defense mechanisms, including TGS and PTGS [17]. Additionally, betasatellites may disrupt chloroplast function, which is crucial for autophagy-mediated degradation of viral proteins [19]. Some non-coding regions of satellite DNA are highly critical for the replication [76], and betasatellite lacking SCR (1190–1320 nt) and its downstream sequences (1091–1190 nt) are essential for replication and symptoms induction [77]. This strategic commandeering

enhances betasatellite amplification and viral movement [83]. Additionally, betasatellites contain iteron-like repeat motif (5'-GAGGACC-3' with 5'-GGACC-3' core), homologous to those in their cognate helper begomoviruses, which are essential for specific Rep binding [82]. Our SCR analyses revealed varying degrees of similarity in conserved repeat motifs across the studied betasatellites. Bβ's motifs exhibited the highest similarity to those in TA, followed by Tbβ. Mβ's motifs showed the least similarity. Notably, Tbβ and Bβ shared a striking proximity in the location of their first motifs, hinting at a possible shared functional significance. These results likely explain the observed performance variations, with Bβ outperforming Mβ despite 96.1% βC1 sequence similarity. This suggests Bβ's motifs are better recognized by Rep than those of Mβ. Collectively, this intricate interplay leads to disease severity, maintenance and trans-replication of betasatellites [31,84].

The stability of the tripartite begomovirus complex under natural *B. tabaci* transmission warrants serial passage experiments. Current field evidence, however, hints at potential stability, as demonstrated by the detection of ToLCNDV with Mβ in field-grown tomato and cotton [31,49]. Our findings further bolster this notion, demonstrating that ToLCNDV can maintain these three betasatellites (Tbβ, Bβ, Mβ), likely facilitated by the *B. tabaci*'s polyphagous nature and capacity to transmit multiple begomovirus components. Nevertheless, without longitudinal co-infection studies spanning multiple plant generations, this claim remains speculative. We acknowledge this as a limitation and propose future serial passage experiments to definitively assess the temporal stability of these associations, enhancing the robustness of our conclusions. It is highly likely that the intensifying agricultural practices in southern Asia will increase the encounter rate between bipartite begomoviruses and betasatellites during co-infections. However, predicting the outcome of these encounters remains elusive, as evidenced by current study and previous research. However, understanding the long-term impact of such interactions will be intriguing. Although betasatellites enhance pathogenicity, host adaptation, suppress RNA silencing, disrupt hormone signaling, causing severe symptoms and expanding the host range. But this dependence on betasatellites may pose evolutionary disadvantages, such as reduced adaptability in their absence, decreased host availability due to increased virulence, and stronger host resistance mechanisms. While betasatellites provide short-term advantages, they may impose long-term trade-offs for ToLCNDV.

While ToLCNDV remains the predominant begomovirus in the region, another bipartite begomovirus, tomato leaf curl Palampur virus (ToLCPalV), also circulates, albeit with a notably less distribution. ToLCPalV is found associated with Pepper leaf curl betasatellite and Mβ [85–87], yet its specific interactions and functional consequences of these associations remain unexplored. This lack of research presents a significant knowledge gap in understanding bipartite begomovirus dynamics. Unlike ToLCNDV, which exhibits widespread adaptability, ToLCPalV's limited range raises intriguing questions about the ecological and molecular factors—such as host specificity, vector efficiency, or Rep-mediated replication dynamics—driving their differential spread and adaptation. Future studies investigating the interactions between ToLCPalV and its associated betasatellites would be valuable in understanding the broader ecological and evolutionary dynamics of begomoviruses in this region. Co-infecting plants with bipartite begomoviruses and betasatellites, potentially through whitefly transmission, and monitoring their persistence over time could address critical questions about their survival dynamics and evolutionary advantages or disadvantages. These queries include: will the betasatellite or the DNA-B ultimately be lost, or could a stable tripartite association evolve? Answering these questions is paramount, as betasatellites are known to exacerbate disease symptoms (and thus losses) and broaden the host range of begomoviruses [26]. Unraveling the answers to these queries can pave way for effective strategies to mitigate the potential risks posed by betasatellite spread and their interaction with bipartite begomoviruses in this rapidly evolving agricultural landscape.

## 5. Conclusions

The interaction between a bipartite begomovirus and a betasatellite was found to be more intricate than just trans-replication of the satellite. Even closely related betasatellites exhibited diverse behaviors, highlighting the complexity of begomovirus-betasatellites interactions. While the apparent antagonism between the betasatellite and DNA-B suggests

a nuanced and potentially unstable relationship, it may not necessarily lead to the formation of super virulent tripartite viruses. However, the regular occurrence of these co-infections in the field underscores their potential to cause substantial economic losses in important crops.

## Supporting information

**S1 File.** **S1 Table.** List of betasatellites found associated with ToLCNDV in Solanaceous crops. **S2 Table.** Oligonucleotide primers used in the study for quantification and Southern blotting. **S3 Table.** Rep-DNA docked complexes and their binding free energy. **S1 Fig.** Diagram showing the positions of primers on ToLCNDV DNA A (A), DNA B (B) and TbLCB (C) that were used in the quantitative PCR assays. **S2 Fig.** Standard curve and melt curve for amplification of ToLCNDV DNA A. (A) Standard curve for estimation of the titre of ToLCNDV DNA A with a PCR efficiency of 96.5%, slope of −3.376 and correlation coefficient 0.992 [left] and a melt curve analysis of the products of amplification [right]. (B) Standard curve for estimation of the titre of ToLCNDV DNA B with a PCR efficiency of 95.0%, slope of −3.376 and correlation coefficient 0.991 [left] and a melt curve analysis of the products of amplification [right]. (C) Standard curve for estimation of the titre of betasatellites with a PCR efficiency of 97.0%, slope of −3.376 and correlation coefficient 0.995 [left] and a melt curve analysis of the products of amplification [right]. The standard curves plot log of starting DNA quantity against threshold cycle [C(T)] cycle. The melt curves plots negative rate of change of fluorescence [-d(RFU)/dT] against temperature. **S3 Fig.** Alignment of coding region βC1 (A) and the satellite conserved region (SCR) (B) from three betasatellites: tobacco leaf curl betasatellite (Tbβ), cotton leaf curl Multan betasatellite (Mβ), and cotton leaf curl Multan betasatellite strain Burewala (Bβ). Sequence similarity is indicated by color: red for highly similar regions across all three betasatellites, green for regions similar in two betasatellites, and blue for dissimilar sequences. Identified repetitive motifs are enclosed in boxes: green for motifs common to all three betasatellites, brown for motifs shared by Tbβ and Bβ, and blue for motifs shared by Bβ and Mβ. Yellow boxes denote previously reported repetitive sequence motifs (Xu et al., 2019). Nonanucleotide sequences are highlighted in red boxes. **S4 Fig.** Quantification of Southern blot bands. Relative band intensities (quantification) from Southern blots are shown for ToLCNDV DNA A (A), single-stranded bands of ToLCNDV DNA A (B), ToLCNDV DNA B (C), and three different betasatellites (D). Band intensities were normalized to the corresponding positive control band within each blot, which was set to a value of 1. Panel (D) illustrates the accumulation of three betasatellites: Mβ (yellow), Bβ (blue), and Tbβ (cyan). **S5 Fig.** Molecular docking models illustrating the predicted binding interactions between the Rep protein and different DNA molecules: (A) Rep protein docked with CR of DNA-A; (B) Rep protein docked with CR of DNA-B; (C) Rep protein docked with SCR of Bβ; (D) Rep protein docked with SCR of Mβ; (E) Rep protein docked with SCR of Tbβ. Docked complexes are shown as representative structures from the HDOCK server, used for protein-nucleic acid docking analysis.
(DOCX)

**S2 File.**
(DOCX)

## Acknowledgments

Authors acknowledge all the support staff in the lab for their assistance during this work.

## Author contributions

**Conceptualization:** Zafar Iqbal, Muhammad Shafiq, Rob W. Briddon.

**Formal analysis:** Zafar Iqbal, Muhammad Shafiq, Sajed Ali, Mudassar Fareed Awan, Muhammad Farhan Sarwar.

**Funding acquisition:** Zafar Iqbal.

**Investigation:** Mudassar Fareed Awan.

**Methodology:** Zafar Iqbal, Muhammad Shafiq.

**Project administration:** Rob W. Briddon.

**Supervision:** Rob W. Briddon.

**Validation:** Zafar Iqbal.

**Writing – original draft:** Zafar Iqbal, Muhammad Shafiq, Imran Amin, Muhammad Shafiq Shahid, Rob W. Briddon.

**Writing – review & editing:** Zafar Iqbal, Imran Amin, Rob W. Briddon.

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
