## [Decision Letter · Decision Letter 0]

Dear Dr. Iqbal,

Thank you for submitting your manuscript to PLOS ONE. After careful consideration, we feel that it has merit but does not fully meet PLOS ONE’s publication criteria as it currently stands. Therefore, we invite you to submit a revised version of the manuscript that addresses the points raised during the review process.

**ACADEMIC EDITOR:**

The study was conducted solely on Nicotiana benthamiana, a model plant that may not fully represent the natural hosts of ToLCNDV and its associated betasatellites. This limits the ecological validity of the findings.

The study does not appear to include field-based experiments, which are necessary to confirm that these interactions occur under natural conditions, especially in cotton-growing regions where these viruses are prevalent. 

Agrobacterium-mediated inoculation does not perfectly mimic natural whitefly-mediated transmission, which could lead to differences in virus accumulation and interactions.

The study relies heavily on qPCR and Southern blot for viral quantification but does not account for potential artifacts like variable DNA extraction efficiency or differential PCR amplification efficiency between different viral components.

While the study discusses interactions at a molecular level, there are no functional assays (e.g., protein interaction studies) to confirm the proposed mechanisms.

The study mentions ANOVA and Tukey’s test for data analysis, but it does not provide details on sample size justification or whether the experiments were adequately powered to detect subtle differences.

The study attributes observed viral titers and symptom severity to interactions between ToLCNDV and different betasatellites but does not explore whether host plant factors (e.g., gene expression differences) contribute to these variations.

The discussion assumes that ToLCNDV acquiring betasatellites enhances pathogenicity and host adaptation, but it does not investigate whether these interactions provide any evolutionary disadvantages to the virus.

The study does not appear to include experiments where betasatellites alone were introduced into N. benthamiana without ToLCNDV to determine their independent effects.

ToLCNDV is studied in isolation with betasatellites, but there is no comparison with other bipartite begomoviruses to determine whether these interactions are unique to ToLCNDV.

The use of Agrobacterium-inoculated infectious clones increases the risk of cross-contamination between viral constructs, potentially leading to misinterpretation of results.

While primer design is mentioned, the study does not report whether primer specificity was validated against non-target viral sequences to rule out nonspecific amplification.

The study does not assess whether ToLCNDV interacting with different betasatellites leads to genetic recombination, which could alter viral evolution and disease dynamics.

Since ToLCNDV is transmitted by whiteflies, the study should have included experiments evaluating whether these viral interactions influence vector-mediated transmission efficiency.

We look forward to receiving your revised manuscript.

Kind regards,

Muhammad Arif, Ph.D.

Academic Editor

PLOS ONE

**Journal Requirements:**

Reviewers' comments:

Reviewer's Responses to Questions

**Comments to the Author**

1. Is the manuscript technically sound, and do the data support the conclusions?

Reviewer #1: Yes

Reviewer #2: Yes

2. Has the statistical analysis been performed appropriately and rigorously?

Reviewer #1: Yes

Reviewer #2: No

3. Have the authors made all data underlying the findings in their manuscript fully available?

Reviewer #1: Yes

Reviewer #2: Yes

4. Is the manuscript presented in an intelligible fashion and written in standard English?

Reviewer #1: Yes

Reviewer #2: Yes

**Reviewer #1:**  The findings highlight the potential emergence of novel and more pathogenic ToLCNDV-betasatellite combinations in cotton-growing regions, stressing their significant implications for disease management and agricultural productivity. However, there are some questions and concerns that need to be addressed. I suggest major revision. Please see the attached file for full comments.

**Reviewer #2: ** This paper studied the interaction between tomato leaf curl New Delhi virus (ToLCNDV) and three betasatellites. They found that the presence of a betasatellite with TA/TB affected the accumulation of ToLCNDV DNA-B and three betasatellites accumulated at different levels with ToLCNDV. These data suggest the intriguing complexity and variability interaction between ToLCNDV and betasatellites. In general, this is a well written paper and the results support the main conclusion. Here, I would like to give few suggestions to further improve the manuscript:

1, the abstract can be shorten.

2, Section 3.3 Quantification of the southern blot is required, as there appears to be a large variation in loading control.

3, Please include all the data points in Figure 6.

4, Discussion should be more focus on the results but not like a review paper.

**Do you want your identity to be public for this peer review?** For information about this choice, including consent withdrawal, please see our Privacy Policy

Reviewer #1: No

Reviewer #2: **Yes: ** Xiaofei Cheng

---

## [Author Response · Author response to Decision Letter 1]

26 Feb 2025

Response to Reviewers

Authors sincerely thank the reviewers and Editor for their insightful and constructive feedback, which has greatly improved the manuscript. Each comment has been addressed in detail (responses in red font, below), and corresponding revisions are marked with track changes in the manuscript. All line numbers referenced pertain to the track-changed version.

Editor and Reviewer Comments:

EDITOR COMMENTS

The study was conducted solely on Nicotiana benthamiana, a model plant that may not fully represent the natural hosts of ToLCNDV and its associated betasatellites. This limits the ecological validity of the findings.

Response: The concern regarding the ecological validity of this study is acknowledged, as Nicotiana benthamiana is a model plant and may not fully represent interactions in natural hosts. However, its use is justified for initial mechanistic insights and controlled experimentation, providing valuable data on virus-host interactions, viral replication, pathogenicity, and host responses. Additionally, while ToLCNDV is a bipartite begomovirus that typically lacks cognate betasatellites, it uniquely demonstrates the ability to associate with and replicate various betasatellites. This study was designed to assess whether ToLCNDV, recently identified from cotton, could interact with prevalent betasatellites in the region and potentially lead to the emergence of new viral combinations that may threaten cotton productivity.

The study does not appear to include field-based experiments, which are necessary to confirm that these interactions occur under natural conditions, especially in cotton-growing regions where these viruses are prevalent.

Response: While we acknowledge the absence of field-based experiments, crucial for validating virus-betasatellite interactions under natural conditions, this study was designed as a controlled laboratory investigation. The motive was to assess if ToLCNDV along with different betasatellites could lead to the emergence of novel combinations. These controlled conditions minimize environmental variability and provide clear insights into viral replication, pathogenicity, and potential recombination, allowing us to establish the mechanistic basis of ToLCNDV-betasatellite interactions. However, we found that while the apparent antagonism between the betasatellite and DNA-B suggests a nuanced and potentially unstable relationship, it may not necessarily lead to the formation of super virulent tripartite viruses.

Agrobacterium-mediated inoculation does not perfectly mimic natural whitefly-mediated transmission, which could lead to differences in virus accumulation and interactions.

Response: The editor's concern regarding the limitations of Agrobacterium-mediated inoculation versus natural whitefly transmission is valid. While Agrobacterium infiltration is a widely utilized and efficient method for delivering viral constructs, it may not fully recapitulate the intricate dynamics of whitefly-mediated transmission, which involves specific vector-virus-plant interactions. However, this method provided the necessary information to address our primary objectives, focusing on the mechanistic basis of viral interactions. We acknowledge that future studies incorporating whitefly transmission would enhance the ecological relevance of our findings.

The study relies heavily on qPCR and Southern blot for viral quantification but does not account for potential artifacts like variable DNA extraction efficiency or differential PCR amplification efficiency between different viral components.

Response: This is a valid point, and we took measures to minimize potential artifacts in our qPCR and Southern blot analyses. To mitigate variability in quantification, the same isolated DNA was used for both Southern blot and qPCR. In addition, in qPCR, different biological and technical repeats were used. Furthermore, qPCR conditions, including primer concentrations and thermocycler settings, were rigorously optimized to ensure consistent amplification efficiency across different viral components. We believe these precautions have minimized potential artifacts. This information has been added to the manuscript at line [L291-298] in the revised version.

While the study discusses interactions at a molecular level, there are no functional assays (e.g., protein interaction studies) to confirm the proposed mechanisms.

Response: Authors appreciate the Editor’s point regarding the lack of functional assays. While protein interaction studies would provide deeper mechanistic insights, they are beyond the scope of this manuscript, which primarily focuses on characterizing ToLCNDV and betasatellite interactions in planta. However, ToLCNDV-encoded Rep binds to iterated repeat motifs on DNA-B and betasatellites to perform replication. We included this point in the revised script by doing some dry analysis (Figure S4B) based on the previously demonstrated study (Xu et al., 2019).

The study mentions ANOVA and Tukey’s test for data analysis, but it does not provide details on sample size justification or whether the experiments were adequately powered to detect subtle differences.

Response: Authors would like to say that everything asked in the query is clearly mentioned in the script. Please see Section 2.7 and legends of the Figure 5.

The study attributes observed viral titers and symptom severity to interactions between ToLCNDV and different betasatellites but does not explore whether host plant factors (e.g., gene expression differences) contribute to these variations.

Response: We acknowledge the editor's point regarding the role of host factors in begomavirus symptom development. It is well-established that begomaviruses induce symptoms like leaf curling, yellowing, and stunting by disrupting host cellular processes through interactions between viral proteins (Rep, TrAP, C4, etc.) and host factors. Differential expression of these host factors significantly influences symptom severity; for instance, upregulation of defense genes can mitigate symptoms, while downregulation of growth-related genes exacerbates them. Given that we observed consistent phenotypic symptoms across three independent experiments, utilizing the same plant genotype, growth conditions, viral load, and Agrobacterium strain, we anticipate that host factor expression remained relatively stable. Therefore, we attribute the observed symptom differences primarily to the variations in viral components. However, a dedicated study specifically investigating host gene expression profiles in response to the different viral components would be necessary to definitively confirm this. We recognize this as a compelling avenue for future research.

The discussion assumes that ToLCNDV acquiring betasatellites enhances pathogenicity and host adaptation, but it does not investigate whether these interactions provide any evolutionary disadvantages to the virus.

Response: This is a very intriguing point which authors missed thoroughly. Considering the importance of this notion, we discussed this in the light of results in the Discussion section, Please see L442-447 in the revised script.

The study does not appear to include experiments where betasatellites alone were introduced into N. benthamiana without ToLCNDV to determine their independent effects.

Response: It is proven by several studies that betasatellites are true satellites that depend on their helper virus for replication, in plants movement, encapsidation and transmission through insect vectors.

ToLCNDV is studied in isolation with betasatellites, but there is no comparison with other bipartite begomoviruses to determine whether these interactions are unique to ToLCNDV.

Response: The other bipartite begomovirus found infecting different plant species is tomato leaf curl Palampur virus (ToLCPalV), which is found associated with pepper leaf curl betasatellite and CLCuMB in the same region but, notably, this virus is not widespread as ToLCNDV. Additionally, no study has investigated such interactions with ToLCPalV but their natural identification from plant hosts. Nonetheless, authors added this notion to the revised script. Please see L448-456 in the revised script.

The use of Agrobacterium-inoculated infectious clones increases the risk of cross-contamination between viral constructs, potentially leading to misinterpretation of results.

Response: While working with the multiple Agrobacterium-inoculated infectious clones could lead to cross-contamination but this wasn’t the case here – as all of our controlled plants remained asymptomatic and didn’t show any PCR-mediated detections.

While primer design is mentioned, the study does not report whether primer specificity was validated against non-target viral sequences to rule out nonspecific amplification.

Response: The used primers were highly specific and yielded only single product – as shown by the melt curve analysis. The same has been mentioned in the revised script (L292-298).

The study does not assess whether ToLCNDV interacting with different betasatellites leads to genetic recombination, which could alter viral evolution and disease dynamics.

Response: This study primarily focused on mechanistic interactions between ToLCNDV and betasatellites, not on assessing recombination. While recombination is a well-documented strategy in begomovirus evolution, it can occur but typically do not occur over just single infection cycles and require sustained co-infections in natural settings. Given the short experimental timeframe, detecting recombination events was beyond this study's scope. However, future longitudinal studies and high-throughput sequencing could help determine whether ToLCNDV-betasatellite interactions drive recombination and impact viral evolution and disease dynamics.

Since ToLCNDV is transmitted by whiteflies, the study should have included experiments evaluating whether these viral interactions influence vector-mediated transmission efficiency.

Response: We acknowledge the editor's point regarding whitefly transmission. While our study focused on in planta interactions between ToLCNDV and betasatellites, driven by their co-occurrence and potential for new viral combinations, vector transmission is crucial. Our primary objective was to establish these in planta interactions, as they are a prerequisite for efficient transmission. Future studies should incorporate whitefly transmission assays to assess vector acquisition and inoculation efficiencies, providing a more comprehensive understanding of these viral interactions.

REVIEWER 1

Major Comments:

1. The study proposes competition between DNA-B and betasatellites for TA-encoded Rep. Was the relative binding affinity of Rep to iterons in DNA-B vs. betasatellites experimentally tested (e.g., EMSA, yeast one-hybrid)? This could directly validate the proposed competition.

Response: Authors appreciate the query as it could provide deeper mechanistic insights into the proposed competition between DNA-B and betasatellites. However, such an investigation is beyond the scope of the current manuscript, which primarily focuses on characterizing the interactions between ToLCNDV and betasatellites in planta. Nonetheless, this is a compelling avenue for future research to directly validate the proposed competition and further elucidate the molecular dynamics governing these interactions.

2. Mβ and Bβ share 96.1% βC1 identity but differ in SCR. How do SCR sequence variations (e.g., non-coding regulatory motifs) influence interactions with ToLCNDV? Were chimeric betasatellites (e.g., swapping SCR regions) tested to isolate SCR-specific effects?

Response: The reviewer raises a valid point about the role of the satellite conserved region (SCR) in modulating interactions with Tomato leaf curl New Delhi virus (ToLCNDV), especially given the high βC1 coding region identity (96.1%) between the Mβ and Bβ betasatellites. While chimeric betasatellites, created by swapping SCR regions, would be an ideal approach to directly isolate SCR-specific effects, this study did not initially pursue that avenue.

Instead, following the reviewer's suggestion, we performed a detailed SCR and repetitive motif analysis across the three betasatellites (Tbβ, Mβ, and Bβ) used in this study. This analysis revealed a greater number of shared repetitive sequence motifs between Tbβ and Bβ compared to those shared between Mβ and Bβ. This difference in shared SCR motifs may explain why Bβ exhibited better performance in our assays compared to Mβ, despite the high βC1 sequence similarity. The SCR contains regulatory elements that influence gene expression, replication, and potentially interactions with the helper virus. The observed differences in motif sharing suggest that the SCR of Bβ may contain elements that are more compatible with ToLCNDV or enhance its function in a way that the Mβ SCR does not. Please see the revised version for these updates; MM (Line 126-133), Results (202-211), and Discussion (426-431).

We acknowledge that the chimeric betasatellite approach would provide more definitive evidence of the SCR's specific role. While not performed in this study, the construction and analysis of such chimeric betasatellites represent a compelling direction for future research.

3. Tbβ, originally from soybean, accumulated more efficiently in N. benthamiana. Does Tbβ encode solanaceous host-specific adaptations (e.g., promoter elements, βC1 interactions)? Were host-switching experiments conducted (e.g., inoculating soybean with ToLCNDV+Tbβ)?

Response: Tbβ was originally isolated from the weed Pedilanthus tithymaloides but has also been reported to co-infect soybean alongside PedLCV (Ilyas et al., 2010). The reviewer's suggestion regarding Tbβ's host range and potential solanaceous adaptations is indeed compelling and offers a promising avenue for future research. While we appreciate the reviewer's insight, exploring these specific adaptations would necessitate a separate, dedicated study. We believe this constitutes substantial research undertaking in itself and, while highly relevant, falls outside the scope of the present work.

4. The conclusion questions whether tripartite associations (ToLCNDV + DNA-B + betasatellite) are stable. Were serial passaging experiments performed to track component persistence over time? If not, how confident are the authors in extrapolating field stability?

Response: The stability of tripartite begomovirus associations (ToLCNDV DNA-A + DNA-B + betasatellite) under natural conditions, particularly with Bemisia tabaci transmission, is a key question, and one we also raised in the Discussion. While serial passaging experiments, which we acknowledge as important, were not performed to directly assess long-term component persistence, existing field evidence supports our assertion of potential stability. The detection of ToLCNDV with CLCuMuB in field-grown cotton (Zaidi et al., 2017) and tomato (Jyothsna et al., 2013) suggests that these tripartite complexes can persist in natural settings. Our finding that ToLCNDV can maintain these three betasatellites (Tbβ, Bβ, and Mβ) further strengthens the likelihood of such stable associations. Furthermore, B. tabaci's polyphagous nature and ability to transmit multiple begomovirus components, along with the natural association of ToLCNDV and CLCuMuB, support this notion. Therefore, while serial passaging experiments would provide more definitive proof, the combined field observations and B. tabaci transmission capabilities suggest these associations can persist long enough for detection in diverse hosts.

We tried to add similar information in the discussion section (Please see L433-438). Hopefully, the kind reviewer will find our explanation satisfactory.

5. Tbβ caused severe symptoms despite lower TB levels. Is this solely due to higher betasatellite titers, or do βC1 functional differences (e.g., suppression of host defenses) play a role? Were βC1 mutants (e.g., deletion/mutation in silencing suppressor domains) tested?

Response: While Tbβ-induced severe symptoms may correlate with its higher betasatellite titers, we cannot rule out potential βC1 functional differences—such as variations in TGS and PTGS—as co

---

## [Decision Letter · Decision Letter 1]

Dear Dr. Iqbal,

Thank you for submitting your manuscript to PLOS ONE. After careful consideration, we feel that it has merit but does not fully meet PLOS ONE’s publication criteria as it currently stands. Therefore, we invite you to submit a revised version of the manuscript that addresses the points raised during the review process.

We look forward to receiving your revised manuscript.

Kind regards,

Rajarshi Gaur

Academic Editor

PLOS ONE

**Comments to the Author**

Reviewer #1: All comments have been addressed

Reviewer #2: All comments have been addressed

2. Is the manuscript technically sound, and do the data support the conclusions?

Reviewer #1: Yes

Reviewer #2: Yes

3. Has the statistical analysis been performed appropriately and rigorously?

Reviewer #1: Yes

Reviewer #2: Yes

4. Have the authors made all data underlying the findings in their manuscript fully available?

Reviewer #1: Yes

Reviewer #2: Yes

5. Is the manuscript presented in an intelligible fashion and written in standard English?

Reviewer #1: Yes

Reviewer #2: Yes

Reviewer #1: The authors addressed my comments. I suggest accepting with revision. The full review with comments can be found in the PDF file.

Reviewer #2: I have carefully check the revision and admit that the authors have answered all my concerns. For the ditor comments, I largely agree with the authors that it is out of the scope to do experiment on natural host, under nature field-based conditions. Therefore, I will suggest a full acceptance this time.

**Do you want your identity to be public for this peer review?** For information about this choice, including consent withdrawal, please see our Privacy Policy

Reviewer #1: No

Reviewer #2: **Yes: ** Xiaofei Cheng

---

## [Author Response · Author response to Decision Letter 2]

8 Apr 2025

Authors sincerely thank the reviewer for his insightful and constructive feedback. Authors tried to address each comment in detail (responses in red font, below), and corresponding revisions are marked with track changes in the manuscript. All line numbers referenced pertain to the track-changed version.

Reviewer Comments:

REVIEWER 1

1. The authors propose that ToLCNDV’s DNA-B and betasatellites compete for the TA-encoded Rep protein, but no direct experimental evidence (e.g., EMSA, yeast one-hybrid, or Co-IP assays) is provided. While they acknowledge this as a limitation, even in silico Rep-binding analysis could help support their hypothesis.

Response: Authors performed Protein-nucleic acid docking studies to verify the hypothesis. All the related sections of the manuscript (MM (L 184-196), Results (L 332-344), and Discussion (L 384-387 and 421-423) have been updated accordingly. All the necessary docking information has been added to the revised Supplementary data.

2. The manuscript suggests that ToLCNDV-betasatellite associations may be unstable over time, but this claim is speculative without long-term co-infection studies. The authors reference field reports of ToLCNDV with CLCuMB but do not experimentally test persistence across multiple plant generations. I suggest the authors acknowledge this limitation explicitly and suggest future longitudinal studies.

Response: Authors acknowledge that this assertion relies on field reports of ToLCNDV co-occurring with CLCuMB without experimental validation of persistence across multiple plant generations. In response, we have explicitly noted this limitation in the revised manuscript (Discussion, L 444-452) and added a recommendation for future longitudinal co-infection studies to assess the stability of these associations over time. These revisions clarify the current evidence gap and outline a path forward to strengthen our findings.

3. The study focuses solely on ToLCNDV interactions, but a comparison to other bipartite begomoviruses (e.g., Tomato leaf curl Palampur virus with betasatellites) would provide a broader perspective. I suggest the authors expand discussion to include known bipartite-betasatellite interactions beyond ToLCNDV.

Response: Authors thank the reviewer for suggesting a broader comparison with other bipartite begomoviruses, such as Tomato leaf curl Palampur virus (ToLCPalV), to contextualize ToLCNDV’s interactions with betasatellites. While our study focuses on ToLCNDV to address its specific prevalence and impact in our sampled regions, we recognize the value of a comparative perspective. In response, we have expanded the Discussion (lines 462–471) to include known bipartite-betasatellite interactions. We appreciate this suggestion and believe it strengthens the manuscript’s contribution to begomovirus research.

---

## [Decision Letter · Decision Letter 2]

Differential interactions of ToLCNDV with different betasatellites reveal complex viral dynamics in N. benthamiana

PONE-D-25-02011R2

Dear Dr. Iqbal,

We’re pleased to inform you that your manuscript has been judged scientifically suitable for publication and will be formally accepted for publication once it meets all outstanding technical requirements.

Kind regards,

Rajarshi Gaur

Academic Editor

PLOS ONE

Additional Editor Comments (optional):

Before final printing author must arrange the references as per the journal norms

Reviewers' comments:

Reviewer's Responses to Questions

**Comments to the Author**

Reviewer #2: All comments have been addressed

Reviewer #3: All comments have been addressed

2. Is the manuscript technically sound, and do the data support the conclusions?

Reviewer #2: Yes

Reviewer #3: Yes

3. Has the statistical analysis been performed appropriately and rigorously?

Reviewer #2: Yes

Reviewer #3: Yes

4. Have the authors made all data underlying the findings in their manuscript fully available?

Reviewer #2: Yes

Reviewer #3: Yes

5. Is the manuscript presented in an intelligible fashion and written in standard English?

Reviewer #2: Yes

Reviewer #3: Yes

Reviewer #2: The authors have answered my concerns, and I have no more concern this time. So, I will suggest a complete acceptance this time.

Reviewer #3: I have critically reviewed the manuscript title “Differential interactions of ToLCNDV with different betasatellites reveal complex viral dynamics in N. benthamiana”, which describes the interaction of begomoviral complex comprising of very devastating bipartite begomovirus Tomato leaf curl New Delhi Virus prevailing Old World begomovirus and three different species of DNA satellite molecules; betasatellites. The manuscript is about very important issue of devastating begomoviruses which belong the largest genus of plant infecting viruses with wide host range. The manuscript analyzed viral pathogenesis utilizing different combinations of begomovirus complex which playing critical role in cotton leaf curl disease epidemics. The manuscript utilized the advanced bioinformatics softwares to analyze the in-silico interaction studies of Rep protein 3D structures with nucleic acid components using molecular docking approach. The manuscript is well written describing the importance of study supported with update review of literature. Methodology of experiments are well explained to reproduce the experimental results. The results and discussions are comprehensively described.

I would recommend the acceptance of manuscript for publication in its present form.

**Do you want your identity to be public for this peer review?** For information about this choice, including consent withdrawal, please see our Privacy Policy

Reviewer #2: **Yes: ** Xiaofei Cheng

Reviewer #3: **Yes: ** Khadim Hussain

---

## [Editor Report · Acceptance letter]

PONE-D-25-02011R2

PLOS ONE

Dear Dr. Iqbal,

I'm pleased to inform you that your manuscript has been deemed suitable for publication in PLOS ONE. Congratulations! Your manuscript is now being handed over to our production team.

Kind regards,

on behalf of

Professor Rajarshi Gaur

Academic Editor

PLOS ONE